# The German Small Lake and Pond Inventory

Alexander Wachholz[1], Susanne I. Schmidt[2], Jens Arle[1], Jeanette Völker[1]

German Environment Agency (UBA), Department for Inland Waters, Wörlitzer Platz 1, 06844, Dessau-Roßlau, Germany

2 Helmholtz Center for Environmental Research (UFZ), Department for Lake Research, Brückstraße 3a, 39114, Magdeburg, Germany

*Correspondence to*: Alexander Wachholz (alexander.wachholz@uba.com)

**Abstract.** Small lakes and ponds are hotspots of biodiversity, biogeochemical reactions, and hydrological interactions in the landscape. While mostly providing the same functions as larger lakes, they often do so at higher rates per unit area.
Exponentially more abundant than larger lakes, small lakes, and ponds contribute significantly to biodiversity, nutrient retention, and the water budget, even at large spatial scales. However, they are rarely considered in regional or larger-scale environmental studies, partly due to a lack of data. To alleviate this, we developed the German Small Lake and Pond Inventory (GSLPI), a comprehensive database of over 260.000 small lakes and ponds ranging in size from 10 to 500.000 m$^2$ (50 ha or 0.5 km$^2$). Using only openly available data from Germany's federal states, we provide information on lake or pond location,
shape, depth, volume, and connectivity. With this database, we aim to facilitate the integration of small lakes and ponds into environmental research and enhance understanding of their roles within changing landscapes.

## 1 Introduction

Lakes play a crucial role in supporting both natural environments and human well-being. While they cover only 2.2 % of the global land area (Pi et al., 2022), their contribution to biodiversity and ecosystem services is disproportionately high (Heino et
al., 2021). Similar to river segments, smaller lakes occur exponentially more frequent than larger ones (Seekell et al., 2013). Here, we defined small lakes as those with mean surface areas < 0.5 km$^2$ (50 ha), as those do not have to be monitored and reported under the European Water Framework Directive ((EU, 2000), Annex II) - they are thus understudied and under-monitored. To further distinguish ponds from small lakes, we used the definition from Richardson et al. (2022), who found that standing water bodies < 5 ha and shallower than 5 meters substantially differ from larger and deeper ones.

While large lakes are responsible for > 90 % of the global lake area (Pi et al., 2022), smaller lakes are exponentially more abundant (Seekell et al., 2013). The role that small lakes and ponds play in their surrounding landscape can hardly be overstated: Per area, smaller lakes and ponds contribute more to biodiversity (Biggs et al., 2017) and the retention of nutrients and sediments (Schmadel et al., 2019) than larger ones. During droughts, lakes and ponds can support their adjacent ecosystems

with water and thereby increase their resilience (Chen et al., 2023). In urban areas, lakes and ponds can significantly contribute to mental health (Völker and Kistemann, 2011) and help to mitigate urban heat island effects by evapotranspirative cooling (Targino et al., 2019). Small lakes and ponds can also be potent emitters of greenhouse gases like $CH_4$ (Pi et al., 2022) and may thus contribute disproportionally to their area to global warming. This is especially true for water bodies used for aquaculture (Rosentreter et al., 2021; Waldemer and Koschorreck, 2023).

However, the exact role that a small lake or pond plays for the adjacent ecosystems depends on its physical characteristics, such as connection to the river network, connection to groundwater, depth, catchment land cover, and its use (Banas et al., 2008; Richardson et al., 2022; Schmadel et al., 2019; Swartz and Miller, 2021). While such characteristics are easy to collect for individual lakes, large sample studies, for example, can be assisted with the availability of such data (Meyer et al., 2024). Furthermore, representative subsets of lakes, e.g. for large-scale sampling campaigns, can be created based on a database of lake characteristics (Leech et al., 2018). An example of such a database is HydroLAKES (Messager et al., 2016) which globally describes lakes > 10 ha and their physical characteristics. A well-known national example is the LAGOS database from the continental United States which contains lakes and ponds > 1 ha (Cheruvelil et al., 2021). LAGOS has been used in a variety of large-scale studies and has been expanded multiple times (Hanly et al., 2024; King et al., 2021; Rodriguez et al., 2023; Stachelek et al., 2022).

On the German national scale, however, no such database exists yet. The only dedicated inventory contains ~800 large lakes and reservoirs (> 50 ha), which are monitored and reported according to the water framework directive (BMUV and UBA, 2022). These lakes are highly affected by eutrophication and only 26 % exhibit a good ecological state according to the water framework directive (Rücker et al., 2019).

Globally available lake databases (e.g. HydroLAKES (Messager et al., 2016), Global Surface Water Explorer (Pekel et al., 2016)) provide data on larger lakes and ponds (> 3 ha) (Ogilvie et al., 2018). Studies on lake size distributions have revealed that the vast majority of lakes are smaller than that (Seekell et al., 2013). Small lakes and ponds may play an especially important role in Germany's landscape. In the glacially formed lowlands of northern Germany, kettle holes can reach densities of up to 40 per $km^2$ (Vyse et al., 2020). In contrast, numerous fishponds are found in eastern Germany (Schwerdtner et al., 2025) and in Bavaria (Lasner et al., 2020), reflecting long-standing anthropogenic influence on small waterbody distribution.

As small lakes are not just ecologically highly relevant, but also seem to be strongly affected by climate and land use change (Pi et al., 2022; Pilla et al., 2020), we consider a dedicated database on the national scale to be a stepping stone towards further understanding and protection of these ecosystems.

To achieve this, we have combined publicly available data from all 16 German federal states to build a national scale database of small lakes and ponds ranging from 1.000 (0.1 ha or 0.001 km²) to 500.000 $m^2$ (50 ha or 0.5 km²), the German Small Lake

and Pond Inventory (GSLPI) which contains more than 262.433 lakes and ponds. In addition to their location and shape, we derived other highly relevant attributes such as relation to the river network, maximum depth and volume, etc., for each lake and pond. For lakes and ponds with a surface area > 1 ha (n=27.210), we also delineated surface catchments. We used the independent and publicly available data set OpenStreetMap to validate the German Small Lake and Pond Inventory.

The GSLPI as well as the python scripts used to generate it are available via a Zenodo repository (Wachholz et al., 2024).

## 2 Data and methods

### 2.1 Lake geometries

The lake polygons, which are the basis of the GSLPI, were obtained from the official cadastral information systems digital landscape model (ATKIS Basis DLM, from here on referred to as DLM) of each federal state. This data is publicly available under the creative common's licenses CC BY 4.0, CC ZERO 2.0, or CC BY 2.0, depending on the federal state (see Table S1 for details of data acquisition and licensing).

From each federal state's DLM, we retrieved all objects of the type 'Standing water bodies' (in German 'AX_Standgewässer'), which are defined as natural or artificial hollow forms of land surface filled with water and enclosed on all sides without any direct connection to the sea (BKG, 2024). In the DLM, inland harbors, which also fulfil this definition, are distinguished from other standing water bodies and were not considered in this work. Single standing water bodies often consist of multiple polygons in the DLM. To identify individual lakes, we added a one-meter buffer to all polygons and then combined all adjacent polygons. The geospatial operations were executed with the Python library GeoPandas (Jordahl et al., 2020) and the code is available together with the data. This resulted in a total of 262.433 individual lakes and ponds ranging from 10 m$^2$ to 500.000 m$^2$ (50 ha or 0.5 km$^2$), which were each given a unique identifier (LakeID). While the federal states DLMs should represent all standing water bodies > 1.000 m$^2$, some states include even small water bodies (Fig. 2). For example, in Bavaria, most derived lake polygons were smaller than 1.000 m$^2$. This heterogeneity between federal states had to be considered when national scale maps or lake density metrics were derived. For this study, we considered, as stated in the DLMs documentation, the lakes and ponds larger than 1.000 m$^2$ (n= 178.194) to be continuously mapped across Germany.

### 2.2 Lake attributes

The attributes we derived for the lakes and ponds are summarized in Table 1.

### 2.2.1 Geometry-related attributes

For each lake and pond, we derived some morphological attributes (area, shoreline length, aspect ratio, shoreline development index) based on their polygons. The shoreline development index (SDI) is defined as

$$SDI = \frac{L}{2\sqrt{\pi A}} \qquad\qquad (1)$$

Where L is the length of the lake's shoreline in meters (circumference of the polygon) and A is its area in square meters. In
principle, the index indicates the complexity of the shorelines' form: a perfect circle has an SDI of 1. More complex forms lead to higher SDIs. As the SDI is scale-dependent, only lakes and ponds of similar size mapped at the same resolution should be compared unless some bias correction is applied (Seekell et al., 2022). The aspect ratio represents the ratio between the lake's longest to its shortest axis. To identify these axes, we fitted rectangles to the lake's polygons' minimum bounding geometries (Cheruvelil et al., 2021). While some lakes are poorly represented by rectangles, the aspect ratio nevertheless helps
to identify elongated forms (high ratio) from square or round forms. We further derived the mean elevation of the lake and pond surfaces from the 10-meter national digital terrain model (BKG, 2016).

### 2.2.2 Connection to other water bodies

The connection between a standing water body and the river network is crucial when trying to understand its function in the surrounding landscape (Schmadel et al., 2019) and the landscape's water budget (Bizhanimanzar et al., 2024).
The river network used here also stems from the ATKIS Basis DLM. River segments with mean widths (during mean flow conditions) < 12 meters are represented as lines with a width class attribute (0-3, 3-6, 6-12 meters, Fig. 1). The location of those lines is given with an accuracy of ± 3 meters (BKG, 2024).  River segments with a mean width greater than 12 meters are represented by polygons. To preserve the topological information, these large river polygons are further characterized by middle lines, which have been estimated by the respective federal authorities. Those middle lines have been supplemented by
fictive connecting lines to connect tributaries or through standing water bodies ((BKG, 2024), Fig. 1).
If the closest river to a lake or pond was wider than 12 meters, we computed the Euclidean distance to the river's segments polygon center (Dist2RunningCenter) and to the rivers' segments' polygons edge (Dist2RunningEdge). If the closest river to a lake or pond was narrower than 12 meters, we computed the distance to the river segment line (Dist2RunningCenter) and reported the width category (RunningWidth) of that segments line (see Fig. 1 for details). This approach allows users of the
database to judge the connectivity between standing water bodies and the river network based on their definitions. We furthermore reported the number of intersections (NRivIntersections) between lake polygon and river networks, as well as the width class of the intersecting river segment (WidthIntersection) and its flow direction (into or out of the lake; FlowDirIntersection).

For the closest distance between standing water bodies edges, we derived two attributes. The first was the Euclidean distance to the next standing water body as listed in the new GLSPI (Dist2Standing). As only standing water bodies > 1000 m$^2$ are continuously mapped in the DLM across the national scale, we calculated a second attribute (Dist2Standing1000) which is the Euclidean distance to the next standing water body ≥ 1000 m$^2$. The first metric cannot be mapped continuously for the entire area of Germany, but it can help, e.g., to identify clusters of lakes. However, since the lakes < 1000 m$^2$ are not mapped

continuously, a Dist2Standing value > x meters cannot be interpreted as proof that there is indeed no standing water body within this range. The second metric (Dist2Standing1000) is only computed for lakes $\geq$ 1000 m$^2$ and only considers neighboring lakes $\geq$ 1000 m$^2$. It is therefore continuous for the entire Germany and can e.g. be used for map making (see Fig. 1). All distance metrics were computed with the sjoin_nearest function of the Python library GeoPandas (Jordahl et al., 2020). We defined the attribute IsInFloodplain to distinguish lakes that are regularly flooded. For this, we used the River flood hazard

maps for Europe and the Mediterranean Basin region (Dottori et al., 2021) with a 10-year return period.

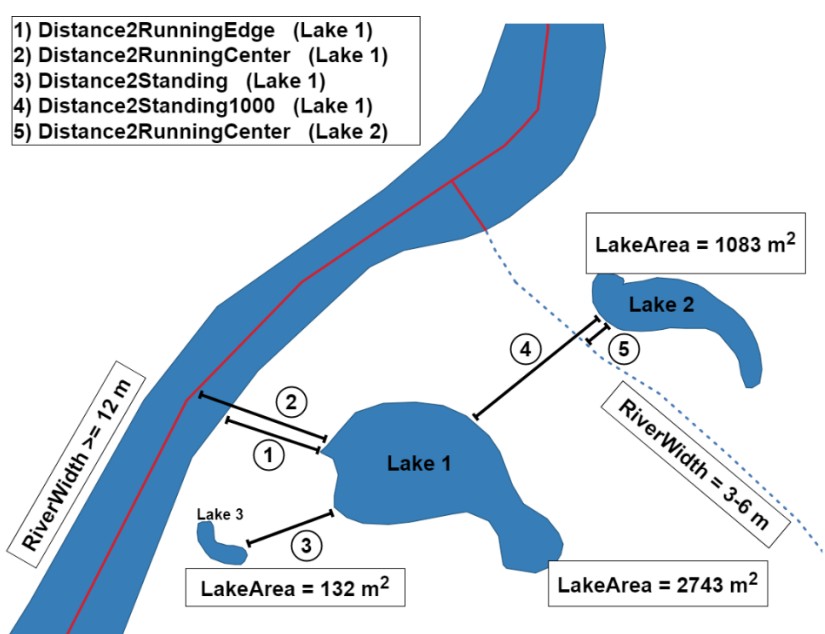

**Figure 1: Schematic representation of river and stream depiction in the ATKIS Basis-DLM (Digital Landscape Model), along with the derived lake–waterbody distance metrics. The blue dashed line indicates a stream or river with a width less than 12 meters, while**
**the blue polygon represents a river wider than 12 meters. The red line illustrates the centerline of the wide river polygon, as well as a hypothetical connection to the smaller stream. If the nearest running waterbody to a lake is a river wider than 12 meters (see Lake 1), both the Distance2RunningEdge and Distance2RunningCenter metrics are calculated. If the nearest running waterbody is narrower than 12 meters, only the Distance2RunningCenter metric is derived. In both cases, the width of the closest river segment is recorded. For standing waterbodies, two separate metrics are calculated: the distance to the nearest standing waterbody**
**(Distance2Standing) and the distance to the nearest standing waterbody larger than 1,000 m². This distinction accounts for the fact that smaller ponds (< 1,000 m²) may be underrepresented in the digital landscape model.**

### 2.2.3 Small lake or pond depth and volume

The depth of lakes and ponds is a key determinant of their ecological functioning. In deeper water bodies, the majority of primary production is more likely to be caused by phytoplankton, while in shallower water bodies submerged or emergent
vegetation can play a major role (Richardson et al., 2022). Lake depth is furthermore a good predictor for other relevant lake attributes, such as mixing regime, timing of the seasonal ice cover, and water quality parameters (Ganz et al., 2024).

We estimated the lake or pond's maximal depth and volume by using the method from (Heathcote et al., 2015), who used the slope of the surrounding landscape. We derived the slope of the surrounding landscape for each lake from the national digital terrain model at 10 meters resolution (BKG, 2016). We validated the results by comparing estimated maximal depths with

observations from ~1.600 bathymetric surveys for lakes provided by the federal state agency for environment, nature conservation and geology of Mecklenburg-Western Pomerania (LUNG) and the Bavarian state agency for the environment (LfU).

## 2.3 Small lake or pond catchments

The composition of a lake's or pond's catchment can provide valuable information on its water quality and ecological state

(Davies et al., 2008; Novikmec et al., 2016). For lakes and ponds with an area larger than 1 ha (10.000 m$^2$, n≈27.200) we derived the surface catchments based on the national digital terrain model at 10 meters resolution (BKG, 2016) using the Python library pysheds (Bartos et al., 2024). For each 10 m x 10 m raster cell from the digital terrain model which was on the lake shoreline, we delineated a catchment and then combined all resulting catchments to get the entire catchment of the lake or pond. We used Corine Land Cover data (EEA, 2018) to estimate how much of the catchment is covered by agricultural,

forest, and urban areas, etc.

Table 1: Short description of all attributes derived for the lake/ ponds, including their storage location in the database.

| Attribute | Description | Unit/ data type | Filename |
|---|---|---|---|
| LakeID | Primary Key for each lake polygon | - | LakeGeometries |
| FedState | Name of the federal state in which the center of the lake or pond is located. | - | LakeGeometries |
| LakeArea | Size of the lake polygon. | [m$^2$] | LakeGeometries |
| LakeShore | Circumference of the lake polygon. | [m] | LakeGeometries |
| SDI | Shoreline development index. | [-] | LakeGeometries |
| AspctRatio | Ratio between the lake's or pond's longest and shortest axis. | [-] | LakeGeometries |
| LakeElevation | Elevation of the lake polygon above sea level. | [m] | LakeAttributes |
| SimZmax | Simulated maximal depth of the lake / pond. | [m] | LakeAttributes |
| SimZmean | Simulated volume of the lake / pond divided by lake / pond area. | [m] | LakeAttributes |
| SimVolume | Simulated volume of the lake / pond. | [m$^3$] | LakeAttributes |
| Dist2RunningEdge | Distance to the edge of the next river. Only if next river is wider than 12 meters. | [m] | LakeAttributes |

| | | | |
|---|---|---|---|
| Dist2RunningCenter | Distance to the center of the next river. | [m] | LakeAttributes |
| WidthRunning | Width class (0-3 m as 3, 3-6 m as 6, 6-12 m as 12, >12 m as 13) of the next river. | [m] | LakeAttributes |
| IsInFloodplain | Is within the flooded area (return period 10 years). | Boolean | LakeAttributes |
| Dist2Standing | Distance to the next lake / pond. | [m] | LakeAttributes |
| Dist2Standing1000 | See above, but only for lakes > 1.000 m$^2$. | [m] | LakeAttributes |
| CatchmentArea | For water bodies > 1 ha (10.000 m$^2$). Size of the catchment polygon. | [m$^2$] | CatchmentGeometries |
| f_clc_lvl3 | Fraction of catchment covered by Corine level 3 land cover class. | [-] | CatchmentLandcover |
| NRivIntersections | Number of intersections between lake / pond and river network. | | LakeAttributes |
| MaxWidthIntersection | Maximal width (see WidthRunning) from intersecting streams/ rivers. | [m] | LakeAttributes |
| WidthIntersection | Width of each river – lake/ pond intersection. | [m] | LakeRivNetIntersectionStats |
| FlowDirIntersection | Flow direction of each river – lake/ pond intersection. | [*in, out*] | LakeRivNetIntersectionStats |

## 2.4 Comparison with small lakes and ponds derived from OpenStreetMaps

The code used to retrieve OpenStreetMap (OSM) data is available together with the data. In our selection process, we queried the OSM API for features classified as ways (features consisting of more than one point) and relations (features consisting of more than one way) of the type "water," explicitly excluding those marked as rivers or streams. We then investigated the "type" of the resulting features and selected 'pond', 'reservoir', 'lake', 'drain', 'oxbow', 'lock', 'fishpond', 'moat', 'natural'. This approach yielded approximately 700,000 water bodies ranging in size from 1 to 500,000 m² which can be found in the Zenodo repository

as well (Wachholz et al., 2024). We then analyzed the overall distribution of lake areas and identified lakes that are mapped in OSM but not included in the GSLPI.

## 3 Results and discussion

### 3.1 Lake geometries

In total, we were able to identify 262.433 small lakes and ponds in Germany, covering 1.2 % of the entire state. 178.194 of

those are larger than 1.000 m$^2$. They show a characteristic distribution across Germany, with high densities occurring in central and northern Bavaria, as well as Mecklenburg-Western Pomerania and Schleswig-Holstein (Fig. 2). High densities of lakes

and ponds in northern Germany are associated with young moraine landscapes while those in northern Bavaria are largely artificial pond landscapes, created since medieval times by localized damming of small rivers for fish farming (Federal Ministry of the Environment, Nature Conservation and Nuclear Safety, 2003). Other prominent areas of high lake and pond densities are the large river valleys of the Elbe and Danube rivers as well as their larger tributaries. These waterbodies are either oxbows formed through natural processes or during river straightening, or excavation lakes that have filled pits created by the mining of riverine deposits (Federal Ministry of the Environment, Nature Conservation and Nuclear Safety, 2003).

While the DLM aims to capture lakes larger than 1.000 $m^2$, some of the federal states report significant numbers of smaller lakes and ponds (Fig. 3). In Bavaria for example, more than 50 % of all reported lakes and ponds are smaller than 1.000 $m^2$. In federal states with high numbers of lakes and ponds < 1.000 $m^2$ (Bavaria, Saxony, and Lower Saxony), we found a correlation between the occurrence of lakes and ponds < 1.000 $m^2$ with those ≥ 1.000 $m^2$. This indicates that, in areas of high lake and pond density, smaller lakes are more likely to be mapped (Fig. A1). Compared to lakes ≥ 500.000 $m^2$, the distribution of smaller lakes is highly heterogeneous across the federal states: in Mecklenburg-Western Pomerania, two thirds of the total lake area are covered by these large lakes. In the Saarland, > 80 % of the total lake area consists of lakes < 500.000 $m^2$ (Fig. 2b).

This should be considered when national maps, e.g. of lake density, are created as lakes < 1.000 $m^2$ are not mapped over the entire area of Germany. The same applies to metrics such as lake-to-lake distances, which will be lower in the federal states that mapped lakes < 1.000 $m^2$.

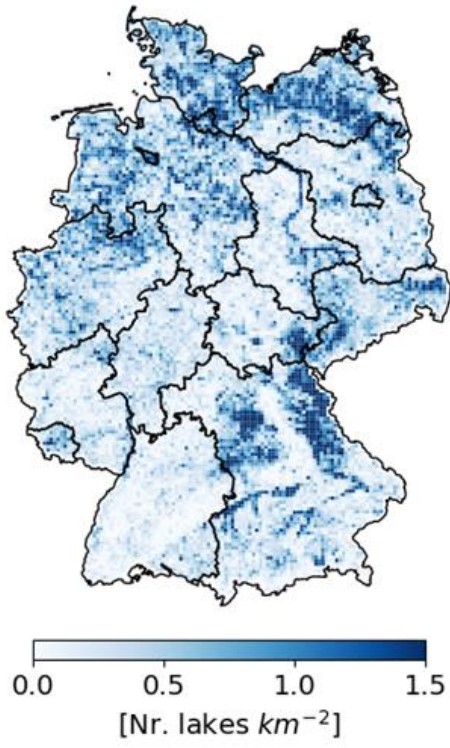


**Figure 2: Density of small lakes and ponds (between 1.000 and 500.000 m²) per square kilometer. A 5 x 5-kilometer grid was used to calculate the density.**

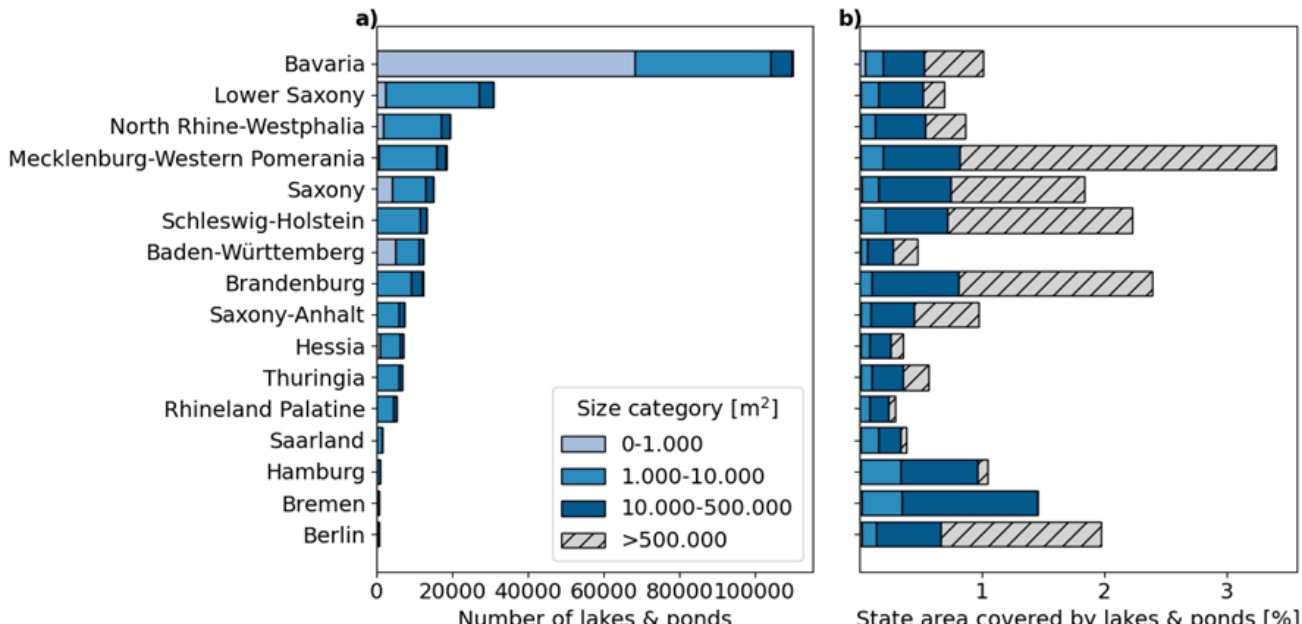

Figure 3: (a) Number and (b) share of federal state area covered by of lakes and ponds of different size classes per federal state. Lakes > 500.000 m$^2$ fall under the reporting obligation of the water framework directive and are not included in the GSLPI. As lakes of the size category > 500.000 m$^2$ are few in numbers, they are not visible in the bar chart of panel a.

## 3.2 Lake attributes

### 3.2.1 Lake geometry attributes

The aspect ratio, which is the ratio between the longest and shortest axis of a geometric object, of lakes and ponds in the GSLPI varies between 1 and 7.3, with a median of 1.8 (Fig. 4b). This indicates that most lakes are slightly elongated. It has to be considered that the aspect ratio is calculated based on the longest and shortest axis of the water bodies minimum bounding geometries (Cheruvelil et al., 2021) and might not be appropriate for more complex geometries.

The shoreline development factor (SDF) ranges from 1 to 8 (data not shown), but should not be interpreted across different water body sizes as it highly correlates with lake or pond area (Seekell et al., 2022).

Lake elevation shows a bi-modal distribution with peaks at 0 and 490 meters above sea level (Fig. 4c). This reflects the high density of lakes in the northern German lowlands, but also in the low mountain ranges of Bavaria, Saxony, and Thuringia.

### 3.2.1 Depth and volume

The simulated maximum depth of the GSLPI ranges from 0 to 170 meters, with a median of 1.5 meters (Fig. 4d). Predicting the depth of a lake from catchment attributes is however highly uncertain (Ganz et al., 2024). Testing our maximum depth, mean depth, and volume predictions against ~1.600 bathymetric maps affirmed this: While the prediction of maximum depth

was only slightly biased (underestimation of 6.1 %, Fig. A2) the root mean square error (RMSE) was 7 meters. Predictions of mean depth and volume significantly underestimated the observed values (50 and 70 % respectively, Fig. A2) with root mean square errors of 4.4 meters and 106 hectometers respectively.

### 3.2.2 Connection to other water bodies

Overall, many lakes and ponds are closely associated with the river network. 32 % of the lakes and ponds are within the positional error (three meters, see Section 2.2.2) of the river network and could therefore have a direct connection to it (Figure 4f, g). Half of the lakes and ponds are within 15 meters of the river network (Figure 4f, g). Most of the lakes and ponds with direct connections to the river network have at least two intersections with the river network (see Fig. 4i) and can be considered "flow-through" lakes. 7 % of all lakes have outflows, but no inflows, and 2 % have in- but not outflows (data not shown). It has to be acknowledged that those categorizations are only feasible for lakes and ponds that are directly connected to the river network. Also, note that lakes that are connected via groundwater will be missed with our methodology. Lakes and ponds within the positional error of the river network but with no connection to it (see Section 2.2.2) cannot be classified. 35 % of all lakes and ponds larger than 1.000 $m^2$ are within 50 meters of each other (Figure 3h). This attribute is likely to be spatially very heterogeneous with low lake-lake distances in Bavaria (Fig. 2). 8 % of all small lakes and ponds are located within active floodplains (defined by areas flooded with a return period of 10 years), covering 7 % of the entire active floodplain area. It should be considered that water bodies in adjacent countries have not been considered, which might lead to an overestimation of the distance to lake and distance to river network metrics in border regions.

### 3.2.3 Catchment area and land cover

Catchment area is expected to increase linearly or exponentially with lake area in logarithmic space (Walter et al., 2020), which is congruent with the findings in this study (Fig. A3).

The land cover of the lake's and pond's catchments is heterogeneously distributed between forest, agriculture, and urban areas etc.: 55 % of the catchments have no urban areas, 38 % no forest areas while only 14 % have no share of agriculture within their catchment (Fig. A4). On average, the catchments are covered by 50 % agricultural areas, 39 % forested areas, 10 % urban areas, 2 % wetlands, and 6 % water bodies.

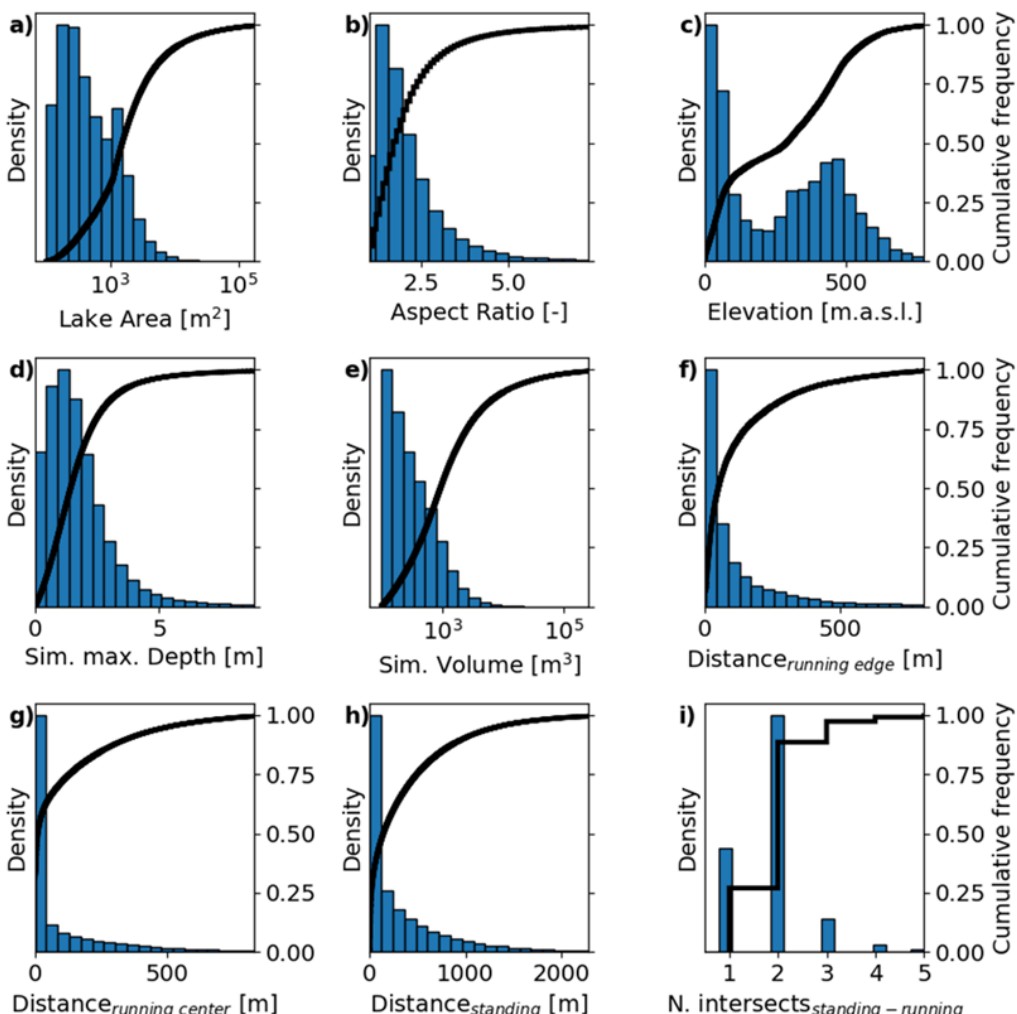

**Figure 4: Normalized histograms (left y axis) and cumulative frequency distributions (right y-axis) for a selection of the derived lake attributes. Each histogram (except panel i) shows the data from the minimum value to the 99th percentile distributed across 20 bins. (a) shows the lake surface area, (b) the lake polygons aspect ratio, (c) its mean elevation above mean sea level, (d) the simulated maximum depth, (e) the simulated volume, (f) the distance to the edge of the closest running waterbody, (g) the distance to the center of the closest running waterbody, (h) the distance to the closest standing waterbody, and (i) the number of intersections between the lake and the river network.**

### 3.3 Comparison with small lakes and ponds derived from OpenStreetMaps

To evaluate the accuracy of lake areas in the federal states' DLM, we compared it with waterbodies collected from OpenStreetMap (OSM). Although OSM is not peer-reviewed, it is compiled independently of the DLM and is thus unaffected by the federal system. This makes it a useful independent reference for identifying potential issues with the GSLPI.

Overall, the GLSPI has ca. 70.000 more lakes in the range of 1.000 – 500.000 m² than OSM (Fig. 5a). 85 % of the OSM lakes can be found in the GSLPI (Fig. 5c, d). The lakes and ponds that exist in both datasets tend to be larger in the GSLPI (Fig. 5b).

This effect seems to decrease with increasing lake and pond size. The recovery rate (% of lakes from OSM found in GSLPI) increases with lake size in a non-linear fashion (Fig. 5c). A possible issue with the collected data could be different mapping practices between the federal states. While we could not exclude this, the fact that in most federal states 85 – 98 % of lakes could be recovered leads us to conclude that different mapping practices could be of minor concern. Only the city-state of Bremen has a lower recovery rate of 55 %, meaning that almost half the lakes and ponds from OSM were absent in the GSLPI.

The fact that the federal state of Bremen only consists of two cities might hint towards an underrepresentation of small lakes and ponds in urban areas. Hamburg and Berlin, the other city-states, have however high recovery rates which indicate that this is a local issue.

    Based on these findings, we consider the GSLPI to be the currently most comprehensive lake inventory for Germany. However, it remains unclear to what extent the small lakes and ponds are permanent. Usually, smaller ponds have a higher probability

of being temporary, especially during droughts (Chumchal et al., 2016).

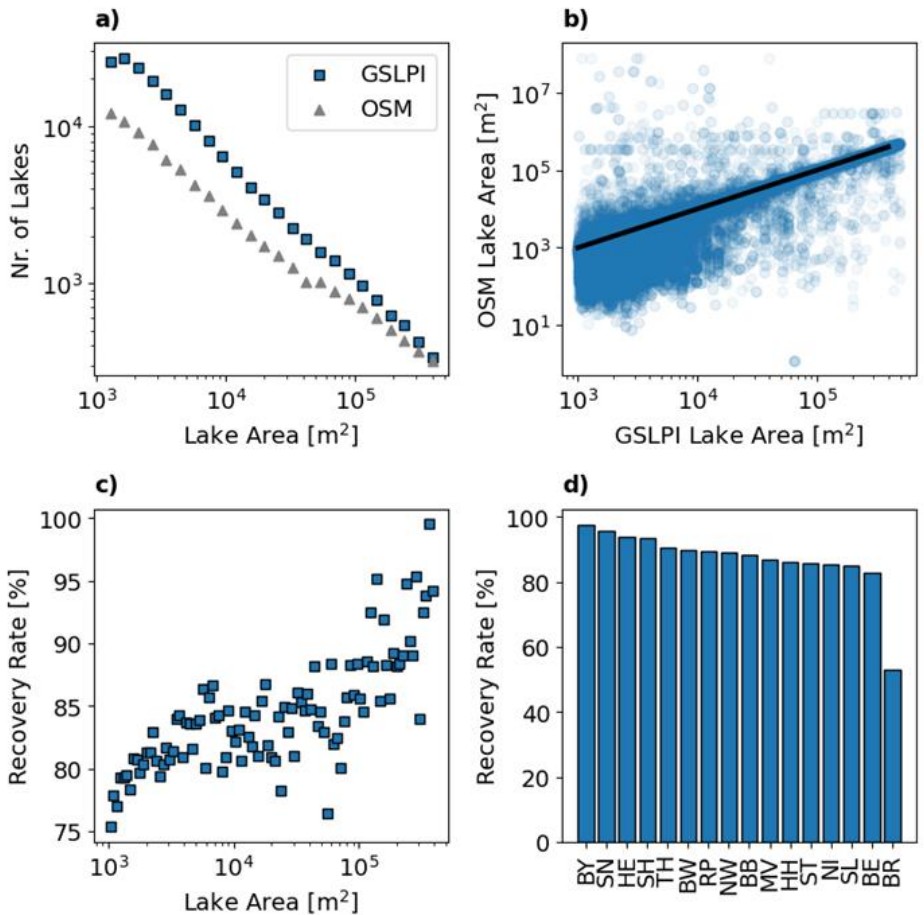

**Figure 5: Results of the comparison between the German small lake and pond inventory (GSLPI) and the standing water bodies retrieved from OpenStreetMap (OSM). (a) shows the log-log size abundance plot for both data sets. In (b) lake surface areas from both data sets are compared. The black line indicates perfect match. The recovery rate (c, d) is the share of lakes from OSM which**

**also exist in the GSLPI shown as a function of lake area (c) and federal state (d).**

## 4. Conclusions

Here we have compiled an open-access inventory of small lakes ($< 500.000$ m², 50 ha or 0.5 km²) and ponds ($< 50.000$ m² or 5 ha) at the national level for Germany. We used other publicly available data sets to estimate attributes such as depth, volume, and relationship to the river network. This data can be used by scientists, policymakers, or practitioners for various purposes.

These can include selecting representative samples of lakes for field studies, improving hydrologic models, or public information. With the increasing availability of high-resolution remote sensing technology, this database can be the foundation of large-scale, low-cost monitoring programs helping to protect these vital ecosystems. Given the critical role small lakes and ponds play in providing ecosystem services and their vulnerability to climate change, this inventory offers crucial support for informed environmental management.

**5. Data and code availability**

The GSLPI as well as the python scripts used to generate it are available in a Zenodo repository at https://doi.org/10.5281/zenodo.14228168 (Wachholz et al., 2024).

**6. Competing interests**

The authors declare that they have no conflict of interest.


**7. Appendix**

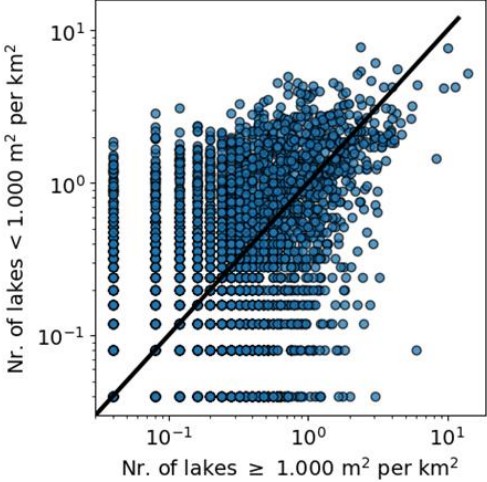

**Figure A1: Relationship between density of lakes ≥ 1.000 and lakes < 1.000 m2. The number of lakes refers to the lakes within each 5 x 5 km grid cell, as shown in Fig.1. The black line represents the 1:1 line.**

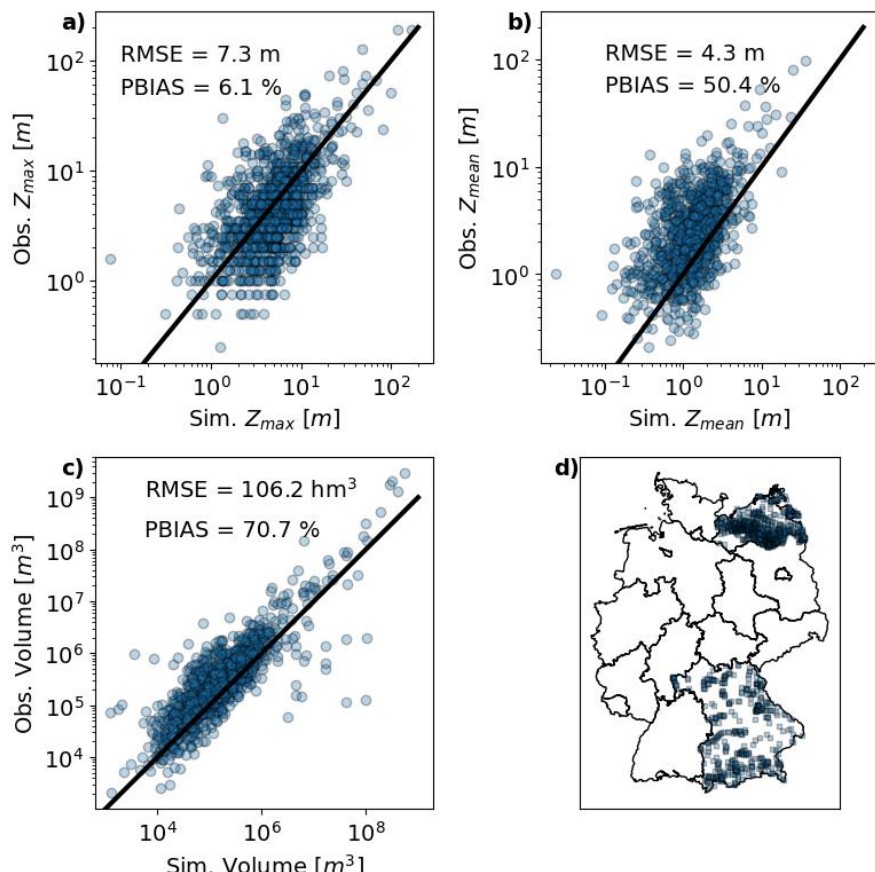


**Figure A2: Simulated versus observed lake or pond maximum depth (a), mean depth (max. depth by area) (b) and volume (c) for ~1.600 lakes and ponds located in Bavaria and Mecklenburg Vorpommern (d). The black lines in panels a, b, c, show the 1:1 lines.**

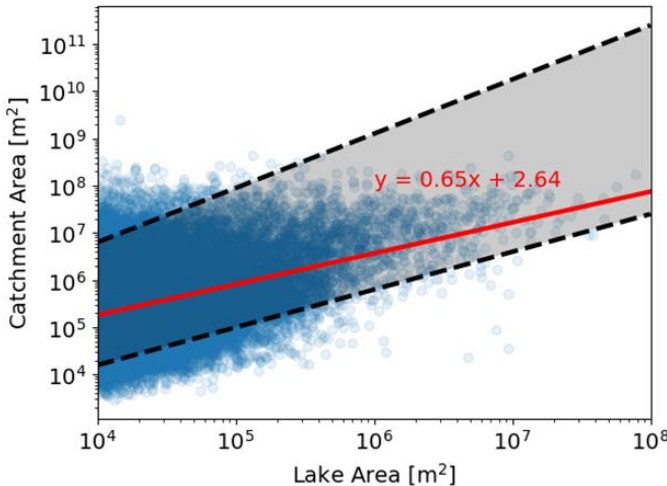

Figure A3: Relationship between lake/ pond and catchment area for all lakes & ponds > 10.000 m2 (1 ha). The red line shows a fitted linear regression line between log10 of lake area and catchment area. The shaded area represents the span of linear relationships between lake and catchment area described by Walter et al. (2020).

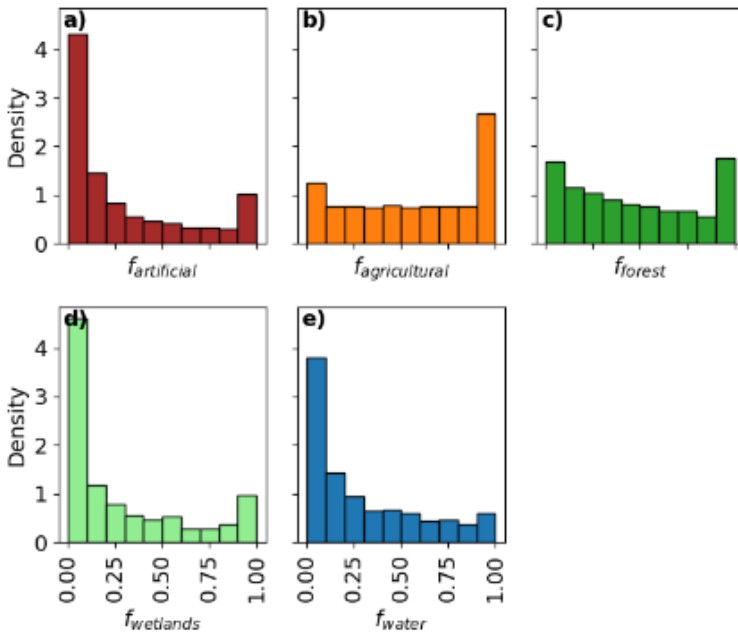

Figure A4: Fraction of the small lake or pond catchments covered by the CORINE level one land cover classes. (a) shows the fraction of artificial surfaces, (b) agricultural areas, (c) forest and semi-natural areas, d) wetlands and (e) water bodies for all catchments.

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
