# Peer review of "The German Small Lake and Pond Inventory"

_Earth System Science Data, 2024_

## Author Response (AR1)

Dear Editors,

we thank both of the reviewers for their detailed comments which we have relied upon to further improve our manuscript. The most substantial changes to the manuscript include the harmonization of figures S1 and S4, which explain crucial terminology to understand the results and are now part of the main manuscript. We further added additional context on German lakes in the introduction and clarified multiple ambiguities throughout the text.

Please find our detailed replies to the reviewer's comments below. As noticed by one reviewer, there were a substantial number of blank lines in the manuscript. Correcting this led to a shift in line numbers. We added the new line numbers (now line xyz) to the reviewers' original comments. In our replies, we also refer to the updated line numbers.

On behalf of the authors,

Alexander Wachholz

**Replies to the comments of reviewer 1:**

Comment 1: The introduction could benefit from additional context regarding the German lakes and ponds.

Reply to comment 1: We agree with the reviewer and have added multiple sentences to this regard (lines 45-55).

Comment 2: From what I understand, the database only covers rivers within Germany's political boundaries. It would be helpful if the dataset also included the extended hydrological catchments that originate in neighboring countries.

Reply to comment 2: We agree with the reviewer that this is a substantial issue which complicates the interpretation of some of the calculated metrics in regions close to the borders. However, no high-resolution river network is yet available on the European scale to our knowledge. We have added a statement reflecting this issue (lines 227 and following).

Comment 3: In the Depth and Volume section, Figure S3 is referenced. I believe this figure should be included in the manuscript.

Reply to comment 3: We understand the reviewers concern and agree that Fig. S3 (now S2) is important. However, to maintain the flow and conciseness of the manuscript, we prefer to keep this figure in the Supplementary Material. This allows interested readers to access the information without interrupting the main narrative. To better convey the information of Fig. S2, we changed the wording regarding the depth and volume estimation in multiple locations (lines 214 & 215, line 267).

Comment 4: Additionally, Figures S1 and S4 could be combined into a single figure and moved from the Supplemental Material to the manuscript.

Reply to comment 4: We agree with the reviewer and have merged figures S1 and S4 to become the new figure 1.

Comment 5: Figures 3 (now Fig. 4) and S6 (now Fig. S4) are missing clarifications for the letters (a, b, c, etc.) in their figure legends.

Reply to comment 5: As suggested, we added the clarifications to the figure captions.

Comment 6: The title of Section 2.4, "Validation", might be reconsidered. While I recognize that OSM is an interesting data source, I wouldn't categorize the procedure as a validation.

Reply to comment 6: We agree with the reviewer and have changed the wording to "Comparison with small lakes and ponds derived from OpenStreetMaps" (lines 159 and 244).

Comment 7: There are two sections (3.2.1 and 3.2.2) both titled "Depth and Volume." This needs to be corrected.

Reply to comment 7: We thank the reviewer for spotting this error and have renamed the section to "Connection to other water bodies" (line 217).

Comment 8: In Section 3.2.2, line 239 (now line 226), it appears that an "enter" key was accidentally pressed, resulting in a second paragraph that doesn't make much sense on its own. This needs to be rewritten for better clarity and understanding.

Reply to comment 8: As suggested the superfluous line break was removed.

Comment 9: "OpenStreetMap" is not used consistently throughout the manuscript. I recommend adopting a consistent format for referring to it. You should choose either "openstreetmap" or "OpenStreetMaps," and I suggest using "OpenStreetMap".

Reply to comment 9: We harmonized the wording to OpenStreetMap.

Comment 10: The reference in line 390 does not follow the correct citation format.

Reply to comment 10: Corrected as suggested. As name and surname were confused originally, the reference now appears in line 285 and following.

Comment 11: There is insufficient space between references in line 445 (Seekell et al. 2022 and Seekell et al. 2013).

Reply to comment 11: Corrected as suggested.

Comment 12: The reference for "Völker et al. 2022" is missing.

Reply to comment 12: We thank the reviewer for spotting this mistake. We have revised the citation in line 45, 46 and added the reference in line 303 and following.

Comment 13: The text formatting needs to be checked, as the manuscript contains many blank spaces, particularly on pages 11, 12, 13, and 14

Reply to comment 13: We thank the reviewer for noticing this and have removed blank spaces on many pages, which lead to substantial shifts in line numbers. We added the new line number as "now line xyz" in parenthesis to the original reviewers comments for orientation.

**Replies to the comments of reviewer 2:**

Comment 1: L. 58, l. 317 (now lines 63, 275): Delete first "d" in "Zendodo repository"

Reply to comment 1: We have corrected the errors as suggested.

Comment 2: L. 69-70 (now lines 73, 74): I do not understand why harbors do not have any direct connection to the sea.

Reply to comment 2: We agree with the reviewer and have clarified that this statement refers to inland harbors (lines 73, 74).

Comment 3: L. 111 (now line 115): Do you mean distance between lake polygon edges or between lake polygon centres?

Reply to comment 3: We thank the reviewer for spotting this ambiguity and have adjusted the wording to clarify (line 115).

Comment 4: Table 1, attribute "SimZMean": Dividing maximal depth by area results in the unit 1/m, not m. That is not an approximation of mean depth, as the term would suggest.

Reply to comment 4: We agree with the reviewer and have corrected the text to indicate that we refer to volume by area, not depth by area.

Comment 5: L. 162 (now line 161): "water ways" rather than "ways"?

Reply to comment 5: We thank the reviewer for spotting this ambiguity. "Ways" and "relations" are here used in the context of the OpenStreetMaps data model. We added short explanations for both terms in lines 161, 162.

Comment 6: L. 165 (now line 164): "_in_ the added reference"

Reply to comment 6: We thank the reviewer for spotting the missing citation and have added it (line 164).

Comment 7: L. 174 (now line 174): Use plural for "other prominent areas"

Reply to comment 7: Corrected as suggested.

Comment 8: L. 181 (now line 181): Unit missing in "< 1.000"

Reply to comment 8: Corrected as suggested.

Comment 9: L. 219 (now line 205): Better "_should_ not be interpreted"

Reply to comment 9: We changed the wording as suggested.

Comment 10: L. 242 (now line 227): Full point missing.

Reply to comment 10: Corrected as suggested.

Comment 11: Figure 4 (now 5) b, figure caption: Replace "perfect correlation" by "perfect match".

Reply to comment 11: We changed the wording as suggested in line 264.

Comment 12: L. 333 (now line 295): The URL for the 10-meter national digital terrain model BKG 2016 https://www.bkg.bund.de/ is now redirecting to https://www.bkg.bund.de/DE/Home/home.html. I guess that more precisely you actually mean https://gdz.bkg.bund.de/index.php/default/digitale-geodaten/digitale-gelandemodelle/digitales-gelandemodell-gitterweite-10-m-dGm10.html, is that right?

Reply to comment 12: We thank the reviewer for spotting this mistake and providing the correct URL, which we used to replace the incorrect one (lines 295, 296).

Comment 13: Figure S2: Do the small polygons denote lakes? It would be very helpful if the different distance measures were indicated in the sketch.

Reply to comment 13: We agree with the reviewer and have updated the figure (see our reply to comment 4 by reviewer 1.).

Comment 14: Figure S2, S3 (now S1, S2): Do the solid black lines denote the 1:1 line?

Reply to comment 14: We added sentences to the captions of figures S2 and S3 which clarify that the lines in deed represent a 1:1 relationship.

Comment 15: Figure S4 is largely redundant to S2 and should be integrated into S2.

Reply to comment 15: We agree with the reviewer. See our reply to comment 4 by reviewer 1.

---

## Author Response (AR2)

Dear Editors,

we thank you for the positive feedback and making us aware that the option of using an appendix exists. As suggested, we moved all the supplementary figures into the appendix (referred to as A) of the main manuscript where they are more accessible. Furthermore, we switched the layout of the only table in the supplement to landscape format to avoid the line breaks and make it more readable.

On behalf of the authors,

Alexander Wachholz

Changes made to the manuscript:

- Line 109: Figure S1 changed to Fig. 1
- Line 182: Fig. S1 changed to Fig. A1
- Line 213, 214: Fig. S2 changed to Fig. A2
- Line 232: Fig. S2 change to Fig. A3
- Line 235: Fig. S4 change to Fig. A4
- Lines 281 and following: Added Appendix Section with the Figures A1 to A4, former S1 to S4 from the supplement